# Influence of Spine Curvature on the Efficacy of Transcutaneous Lumbar Spinal Cord Stimulation

**DOI:** 10.3390/jcm10235543

**Published:** 2021-11-26

**Authors:** Veronika E. Binder, Ursula S. Hofstoetter, Anna Rienmüller, Zoltán Száva, Matthias J. Krenn, Karen Minassian, Simon M. Danner

**Affiliations:** 1Institute for Analysis and Scientific Computing, Vienna University of Technology, 1040 Vienna, Austria; VeronikaBinder@gmx.at; 2Center for Medical Physics and Biomedical Engineering, Medical University of Vienna, 1090 Vienna, Austria; karen.minassian@meduniwien.ac.at; 3Department of Orthopedic and Trauma Surgery, Medical University of Vienna, 1090 Vienna, Austria; anna.rienmueller@meduniwien.ac.at; 4Competence Center Medizintechnik, Austrian Workers’ Compensation Board, 1200 Vienna, Austria; z.szava@gmail.com; 5Department of Neurobiology and Anatomical Sciences, University of Mississippi Medical Center, Jackson, MS 39216, USA; mkrenn@umc.edu; 6Center for Neuroscience and Neurological Recovery, Methodist Rehabilitation Center, Jackson, MS 39216, USA; 7Department of Neurobiology and Anatomy, College of Medicine, Drexel University, Philadelphia, PA 19129, USA

**Keywords:** biophysics, H reflex, human, M wave, neuromodulation, posterior root-muscle reflex, posterior root stimulation, spine alignment, spinal cord, spinal cord stimulation, spine, transcutaneous

## Abstract

Transcutaneous spinal cord stimulation is a non-invasive method for neuromodulation of sensorimotor function. Its main mechanism of action results from the activation of afferent fibers in the posterior roots—the same structures as targeted by epidural stimulation. Here, we investigated the influence of sagittal spine alignment on the capacity of the surface-electrode-based stimulation to activate these neural structures. We evaluated electromyographic responses evoked in the lower limbs of ten healthy individuals during extension, flexion, and neutral alignment of the thoracolumbar spine. To control for position-specific effects, stimulation in these spine alignment conditions was performed in four different body positions. In comparison to neutral and extended spine alignment, flexion of the spine resulted in a strong reduction of the response amplitudes. There was no such effect on tibial-nerve evoked H reflexes. Further, there was a reduction of post-activation depression of the responses to transcutaneous spinal cord stimulation evoked in spinal flexion. Thus, afferent fibers were reliably activated with neutral and extended spine alignment. Spinal flexion, however, reduced the capacity of the stimulation to activate afferent fibers and led to the co-activation of motor fibers in the anterior roots. This change of action was due to biophysical rather than neurophysiological influences. We recommend applying transcutaneous spinal cord stimulation in body positions that allow individuals to maintain a neutral or extended spine.

## 1. Introduction

Transcutaneous lumbar spinal cord stimulation was designed to activate large-diameter posterior root afferent fibers through the use of skin-surface stimulation electrodes placed over the spine at the thoracolumbar junction, overlying the lumbosacral spinal cord, and indifferent electrodes over the anterior lower trunk [1]. Single stimuli were shown to evoke posterior root-muscle (PRM) reflexes (short-latency spinal reflexes with physiological similarities to the soleus H reflex) in many lower limb muscles [1,2,3,4,5]. Trains of stimuli can therefore provide tonic multisegmental afferent input to the spinal cord, comparable with epidural spinal cord stimulation [6]. As a result, transcutaneous lumbar spinal cord stimulation has been used for neurophysiological studies by investigating reflex modulation in multiple lower limb muscles simultaneously [7,8,9] as well as for neuromodulation of sensorimotor function after spinal cord injury [10,11,12] and multiple sclerosis [13,14].

To reliably apply transcutaneous lumbar spinal cord stimulation, posterior root afferent fibers must be activated selectively and consistently. Although the stimulation seems unspecific, the biophysical properties of the lumbosacral spinal cord and its surrounding spinal roots enable a rather selective recruitment of large-diameter posterior root fibers [15,16,17,18]. Yet, under certain conditions, anterior roots containing the axons of the spinal motoneurons can be (co-)activated [15,16]. In neurophysiological studies, this (co-)activation would evoke direct M wave-like responses [1,2] that bypass the spinal sensorimotor circuits; in neuromodulation approaches, continuous contractions of the respectively innervated muscles would be generated, impeding the intended effects. We have previously shown that selective posterior root recruitment can be achieved when stimulation is applied in supine or normal upright standing positions [1,3,6,10,19], but when applied in the prone position, the thresholds of anterior root motor fibers relative to those of the posterior roots are decreased [19].

Which neural structures are being activated depends on the interplay of several biophysical and anatomical factors that influence the electric field acting on the target neural structures [15,16,17,18,19]. The electric current crosses the spine mainly through the soft tissues between the bony vertebrae [15,20]. With flexion and extension of the spine, the vertebrae undergo relative movements and rotations, resulting in changes in the distance between bony structures, in the dimensions of the intervertebral foramina, and in deformation of the soft tissues [21].

We here hypothesized that these changes in the properties of the volume conductor in-between the surface electrodes would influence the current flow through the soft tissues of the spine and, in turn, the current acting on the anterior and posterior root fibers. We applied transcutaneous spinal cord stimulation in individuals with intact nervous systems who assumed three spine alignment conditions (neutral, extension, flexion), each within four body positions (supine, lateral recumbent, sitting, standing). We show that PRM reflexes of similar size are elicited in the extended and neutral spine alignment condition, while they diminish in the flexed spine alignment condition and co-activation of motor fibers in the anterior roots occurs. By concomitantly monitoring the H reflex, we demonstrate that these effects are not caused by neurophysiological modulation but rather by biophysical influences on the stimulation conditions. Such influences are important to consider because a neutral spinal curvature cannot always be maintained by individuals with sensorimotor disorders, and spinal curvature can change during dynamic motor tasks.

## 2. Materials and Methods

### 2.1. Participants

Ten neurologically intact individuals (21–33 years; three females; body mass index: 18.8–25.7, mean: 21.8) participated in the study. Exclusion criteria were previous spinal, cranial, or abdominal surgeries, meningitis, primary muscular diseases, pronounced postural anomalies or restricted flexibility of the lower limbs, and pregnancy. The study was approved by the institutional review board of the Medical University of Vienna, Austria (EK 1686/2014), and conducted in accordance with the Declaration of Helsinki. All participants signed written informed consent prior to their enrollment into the study.

### 2.2. Investigated Body Positions and Spine Alignments

Lower-limb muscle responses to tibial nerve stimulation and transcutaneous spinal cord stimulation were studied in the supine, lateral recumbent, sitting, and standing positions. In each of these body positions, three static alignments of the thoracolumbar spine in the sagittal plane were studied: maximum extension, neutral, and maximum flexion. Maximum extension and flexion were defined as the maximum displacements that the participants felt comfortable to assume. Maximum extension, neutral, and maximum flexion angles were not controlled to be the same across body positions. The experiments were designed to investigate the effects of relative changes in spine alignment. The spine was straight across positions in the frontal plane, with the head and neck aligned in a neutral position. In the supine and standing positions, the arms were relaxed, parallel to the torso; in the lateral recumbent position, the elbows were flexed with the forearms in a comfortable position. In the sitting position, the forearms rested on the thighs. In the supine position, the examination table was adjusted, and therapeutic pillows were used to support and stabilize the different spine alignments. In the lateral recumbent position, participants lay on their left sides, a support ensured a straight spine in the frontal plane, and stabilization was provided to support the pelvis and shoulder girdle against rotations. In the sitting position, the participants sat on the examination table with their feet placed on a height-adjusted footrest and with hip, knee, and ankle flexed at 90°. In the standing position, the participants assumed an upright standing position next to a wall for reference, with no external support and with the shoulders aligned above the hips across spine alignment conditions. An orthopedic surgeon (A.R.) monitored proper positioning of the study participants throughout the recordings. An ankle orthosis (Malleo Sprint, Otto Bock HealthCare GmbH, Duderstadt, Germany) was used to maintain equal muscle length of the left soleus across recordings.

### 2.3. Electromyographic Recordings

Surface electromyographic (EMG) activity was acquired bilaterally from rectus femoris, the hamstrings muscle group, tibialis anterior, and soleus with pairs of silver-silver chloride electrodes (Intec Medizintechnik GmbH, Klagenfurt, Austria) placed with an interelectrode distance of 3 cm (Figure 1) [6,22]. Ground electrodes were placed over the lateral malleolus of each leg. An abrasive paste (Nuprep, Weaver and Company, Aurora, CO, USA) was used to reduce EMG electrode resistance below 10 kΩ. EMG signals were amplified with a gain set to 602, filtered to a bandwidth of 10–600 Hz, digitized at 10,000 samples per second and channel using a USB-NI 6261 data acquisition card (National Instruments, Inc., Austin, TX, USA), and recorded using DasyLab 12.0 (Measurement Computing Corporation, Norton, MA, USA).

### 2.4. Stimulation Procedures

For all stimulation procedures, a current-controlled stimulator (Stimulette r2x-S1, Dr. Schuhfried Medizintechnik GmbH, Moedling, Austria) was used and set to deliver charge-balanced, symmetrical, biphasic rectangular pulses of 1 ms width per phase.

Stimulation of the posterior tibial nerve of the left leg was carried out using a self-adhesive hydrogel surface electrode (Ø = 1 cm; Leonhard Lang GmbH, Innsbruck, Austria) placed in the popliteal fossa and a reference electrode (Ø = 5 cm; Schwamedico GmbH, Ehringhausen, Germany) placed over the anterior aspect of the knee (Figure 1i). The electrode position in the popliteal fossa was adjusted to minimize the stimulation amplitude required to elicit an H reflex in the soleus while producing an isolated ankle plantar-flexion movement when increasing the stimulation amplitude. Maximum M waves (Mmax) were determined with the participants lying supine with neutral spine alignment.

The H reflex was used to assess potential neurophysiological influences of the different static spine alignment conditions on reflex excitability. To this end, tibial-nerve stimulation amplitudes were first separately adjusted for a neutral spine alignment in each of the four body positions to elicit H reflexes with peak-to-peak amplitudes corresponding to 25% of Mmax on the ascending limb of the recruitment curve [23,24] with present M waves. For each body position, the stimulation amplitude was then kept constant across the extension, neutral, and flexion conditions. With the designated stimulation amplitude, six stimulation pulses were applied per body position and spine alignment condition, separated by a minimum of 8 s between consecutive stimuli.

Stimulation of the lumbar and upper sacral spinal roots was carried out through a self-adhesive hydrogel surface electrode (Ø = 5 cm; Schwamedico GmbH, Ehringhausen, Germany) placed medially over the spine between the T11 and T12 spinous processes and a pair of interconnected electrodes (each 8 × 13 cm) on the lower abdomen (Figure 1ii). With reference to the abdominal electrodes, the paraspinal electrode acted as the anode for the first phase of the biphasic stimulation pulses and as the cathode for the second phase [6].

To study the dependence of transcutaneous lumbar spinal cord stimulation on spine alignment, stimulation amplitudes were adjusted in each body position with a neutral spine alignment to elicit EMG responses in the left soleus with peak-to-peak amplitudes that best matched those of the H reflexes in this muscle as well as to concomitantly elicit responses in all other muscles studied. For each body position, stimulation amplitudes were kept constant across the extension, neutral, and flexion conditions. With the designated stimulation amplitude, three single stimulation pulses and three double stimulation pulses with an interstimulus interval of 35 ms were applied, separated by a minimum of 8 s between consecutive single or double stimuli. Correct adhesion of all stimulation electrodes was regularly checked in each of the positions and spine alignment conditions.

### 2.5. Study Protocol

Each subject assumed body positions in the following order: supine, lateral recumbent, standing, and lastly, sitting. For each body position, the stimulation intensities for tibial nerve stimulation and transcutaneous lumbar spinal cord stimulation were first adjusted (as described above) with neutral spine alignment. Then, both types of stimulation were applied in neutral, followed by extended and flexed spine alignment. For each body position and spine alignment condition, first tibial nerve stimulation and then transcutaneous spinal cord stimulation was applied.

### 2.6. Data Analysis

Before the analysis, all recordings were visually inspected. For each body position and spine alignment condition investigated, EMG peak-to-peak amplitudes were calculated for the H reflexes and M waves elicited in the left soleus. Peak-to-peak amplitudes were also calculated for EMG responses elicited by single-pulse and paired-pulse transcutaneous spinal cord stimulation in rectus femoris, hamstrings, tibialis anterior, and soleus of both legs. For the responses to double-pulse transcutaneous spinal cord stimulation, the peak-to-peak amplitudes of the responses to the second stimulus of the pair were normalized to the respective first ones. Normalized second-to-first response amplitudes were only included in the statistical model if the corresponding first response sizes were > 100 µV.

Statistical analysis was performed with R 4.0.0 (R Foundation for Statistical Computing, Vienna, Austria) [25]. Each statistical model included a full factorial dispersion model, fit using Template Model Builder [26,27] interfaced through the glmmTMB package [28]. Backward elimination was performed on the random effects. Random effects were removed from the model if the likelihood-ratio test was not significant. Model assumptions were tested using the DHARMa package for R with distribution, dispersion, outliers, and quantile deviation tests performed. Q-Q plots and plots of residuals against the predicted values were inspected. If any assumptions were violated, other error distributions and link functions were tested. To control for multiple comparisons, Bonferroni–Holm correction was applied to all post hoc tests [29]. An alpha error of *p* < 0.05 was regarded as significant.

Separate generalized linear mixed models with spine alignment condition (extended, neutral, flexed) as fixed effect and body position (supine, lateral recumbent, sitting, standing) as random effect were run for the EMG peak-to-peak amplitudes of H reflexes, M waves, and responses evoked by transcutaneous spinal cord stimulation in the left soleus. For the statistical model of the M wave, a Gamma distribution with a log link function was used to satisfy model assumptions.

For the responses to transcutaneous lumbar spinal cord stimulation in rectus femoris, hamstrings, tibialis anterior, and soleus, results derived from the left and right lower limbs were considered as separate cases. Separate generalized linear mixed models with spine alignment condition, muscle group, and their interaction as fixed effects and body position as a random effect were performed for the EMG peak-to-peak amplitudes of responses elicited by single (or respective first stimulation pulses of a pair) as well as for the normalized second-to-first response amplitudes. A Gamma distribution with a log link function was used for both models to ensure all model assumptions were met.

Recordings of one individual in the lateral recumbent and sitting body positions were excluded from analysis because the subject was not able to hold these positions steadily without generating EMG activity in the lower limb muscles.

## 3. Results

### 3.1. Influence of Spine Alignment Condition on H Reflexes and PRM Reflexes in Soleus

Soleus EMG responses evoked by tibial nerve stimulation as well as by transcutaneous spinal cord stimulation in an individual in a lateral recumbent position with extension, neutral alignment, and flexion of the thoracolumbar spine are displayed in Figure 2. M waves and H reflexes with similar EMG peak-to-peak amplitudes were evoked across spine alignment conditions (Figure 2A). In the extension and neutral spine alignment conditions, transcutaneous spinal cord stimulation evoked PRM reflexes—as documented by the post-stimulation depression of the second responses when double-stimuli were applied [1,6] (Figure 2B). The EMG responses were comparable in the extension and neutral spine alignment conditions but diminished in size when stimulation with the same amplitude was applied in the flexed condition. This reduction of the response size with spinal flexion was likely caused by biophysical influences because no depression of the H reflex was observed (Figure 2A).

The generalized linear mixed model on the peak-to-peak amplitude of the soleus H reflex revealed a significant fixed effect of spine alignment condition on the attained peak-to-peak amplitudes (F_2;97_ = 4.355, *p* = 0.0154, ${ŋ}_{p}^{2}$ = 0.082; Figure 3A). Post hoc tests demonstrated larger peak-to-peak amplitudes with flexed compared to neutral spine conditions (t_97_ = 2.859, *p* = 0.0143), with no further significant differences detected (extension vs. flexion, t_97_ = 2.275, *p* = 0.0642; extension vs. neutral alignment, t_97_ = 0.655, *p* = 0.7902). Further in-depth post hoc analyses unveiled significantly larger peak-to-peak amplitudes with flexed compared to neutral spine alignment only in the lateral recumbent position (t_88_ = −2.805, *p* = 0.0169; all other *p* > 0.100). In the case of the M wave, no significant effect of spine alignment condition on the attainable peak-to-peak amplitudes was detected (F_2;105_ = 1.816, *p* = 0.1678, ${ŋ}_{p}^{2}$ < 0.0001), signifying consistent tibial nerve stimulation conditions.

The statistical model for the soleus peak-to-peak amplitudes of the EMG responses to transcutaneous spinal cord stimulation revealed a highly significant fixed effect of spine alignment condition on the response size (F_2;98_ = 47.879, *p* < 0.0001, ${ŋ}_{p}^{2}$ = 0.494; Figure 3B). Post hoc tests revealed significantly smaller peak-to-peak amplitudes with flexed compared to the neutral (t_98_ = 9.410, *p* < 0.0001), as well as to the extended spine alignment conditions (t_98_ = 7.335, *p* < 0.0001). No differences were found between extended and neutral spine alignments (t_98_ = 0.781, *p* = 0.7153).

### 3.2. Influence of Spine Alignment Condition on Responses in Thigh and Leg Muscles Evoked by Transcutaneous Spinal Cord Stimulation

EMG responses of multiple lower-limb muscles to transcutaneous spinal cord stimulation elicited in an individual standing upright and with extension, neutral alignment, and flexion of the thoracolumbar spine are displayed in Figure 4. Stimulation applied in the spinal extension and neutral spine alignment conditions evoked PRM reflexes in hamstrings, tibialis anterior, and soleus, as demonstrated by the post-activation depression of the second responses when double-stimuli were applied. With flexion of the spine, however, response sizes diminished in these muscles, despite the unchanged stimulation amplitude. In rectus femoris, double stimuli evoked two responses of similar EMG peak-to-peak amplitudes, revealing direct M wave-like response-components elicited by anterior root stimulation [2,15]. Indeed, the shape of the EMG responses in rectus femoris to the first and second pulses of the double stimuli indicate mixed stimulation of upper lumbar posterior and anterior roots with spine extension and neutral spine alignments, but only anterior root (without posterior root) stimulation with spine flexion (Appendix A Figure A1). Direct anterior root recruitment can be the dominating effect of stimulation when applied in a sitting position (Figure A2).

The statistical model run for the responses to the single or first stimulation pulses of a pair revealed a significant fixed-effect of the spine alignment condition on the attainable EMG peak-to-peak amplitudes of responses across muscles (F_2;720_ = 23.660, *p* < 0.0001, ${ŋ}_{p}^{2}$ = 0.062) with smaller responses evoked with flexed than extended (t_720_ = 5.691, *p* < 0.0001) and neutral spine alignments (t_720_ = 6.831, *p* < 0.0001; Figure 5i). No differences in response size were found between extended and neutral spine alignments (t_720_ = 1.257, *p* = 0.4200). Across spine alignment conditions, the fixed effect of muscle was significant (F_3;720_ = 187.233, *p* < 0.0001, ${ŋ}_{p}^{2}$ = 0.438), indicating that response amplitudes differed between muscles (Figure 5ii). All post hoc pairwise comparisons were significant (all *p* < 0.001). EMG peak-to-peak amplitudes were smallest in rectus femoris and largest in soleus. Finally, the interaction between spine alignment condition and muscle was significant, (F_6;730_ = 3.193, *p* = 0.0040, ${ŋ}_{p}^{2}$ = 0.026), indicating that the influence of spine alignment on the stimulation differed between muscles (Figure 5iii). Planned pairwise contrasts revealed significantly lower EMG peak-to-peak amplitudes of the responses in hamstrings, tibialis anterior, and soleus when evoked with flexed compared to both extended as well as neutral spine alignments (all *p* < 0.05). No such differences were found in the case of rectus femoris (extended vs. flexed spine, *p* = 0.9693; neutral vs. flexed spine alignment, *p* = 0.9762).

Double stimuli applied with an interstimulus interval of 35 ms provided essential information on the influence of spine alignment condition on the recruitment of afferent fibers in the posterior roots and potential concomitant activation of motor fibers in the anterior roots—larger normalized second-to-first response amplitudes without changes in voluntary descending inputs indicate an increased likelihood of motor fiber co-activation [1,2,19]. The statistical models run for the normalized second-to-first response amplitudes revealed a significant effect of the spine alignment condition across muscles (F_2;569_ = 61.288, *p* < 0.0001, ${ŋ}_{p}^{2}$ = 0.177), with significantly larger normalized second-to-first response amplitudes obtained with flexed than with extended (t_569_ = 8.886, *p* < 0.0001) or neutral spine alignments (t_569_ = 10.551, *p* < 0.0001; Figure 6i). No differences were found between extended and neutral spine alignments (t_569_ = 1.539, *p* = 0.2736). Across spine alignment conditions, the fixed effect of muscle was significant (F_3;569_ = 125.374, *p* < 0.0001, ${ŋ}_{p}^{2}$ = 0.398), with the largest normalized second-to-first response amplitudes found in rectus femoris and the smallest in soleus (Figure 6ii). All pairwise post hoc comparisons were significant (all *p* < 0.0001), except for the one between hamstrings and tibialis anterior (*p* = 0.1631). Finally, the interaction between spine alignment condition and muscle was significant (F_6;569_ = 8.759, *p* = 0.0040, ${ŋ}_{p}^{2}$ = 0.085), indicating that the effect of spine alignment on normalized second-to-first response amplitudes differed between muscles (Figure 6iii). Pairwise contrasts revealed significantly larger normalized second-to-first response amplitudes in hamstrings, tibialis anterior, and soleus when evoked with flexed compared to both extended as well as neutral spine alignments (all *p* < 0.002). No such differences were found in the case of rectus femoris (extended vs. flexed, *p* = 0.3276; neutral vs. flexed, *p* = 0.2423).

## 4. Discussion

We demonstrated that the alignment of the spine in the sagittal plane affects the capacity of transcutaneous spinal cord stimulation to recruit posterior root afferents. While PRM reflexes could be elicited in neutral and extended spine alignment conditions, flexed spine alignment resulted in a strong reduction of the response amplitudes and in (co-)activation of motor fibers. The underlying mechanisms were of biophysical rather than neurophysiological nature as suggested by the lack of depression of the tibial nerve-evoked soleus H reflex in the same condition. To improve the generalizability of the results, we performed the study in four different body positions and controlled for position-specific effects in the statistical models.

Similarly to epidural electrical stimulation [30,31,32,33], the target neural structures of transcutaneous spinal cord stimulation are afferent fibers within the posterior rootlets of several spinal cord segments, as corroborated by computer simulations [15,17,18,34] and neurophysiological studies [1,6,35]. Additionally, computer simulations of transcutaneous spinal cord stimulation identified low-threshold sites of motor axons within the anterior roots at their exits from the vertebral canal in the intervertebral foramina, where they join the corresponding posterior roots to form the spinal nerves [15,17]. Stimulation at such sites would lead to a mixed activation of afferent and efferent axons.

Mixed activation of sensory posterior root and motor anterior root fibers would be disadvantageous for both major applications of transcutaneous spinal cord stimulation, i.e., neurophysiological and interventional studies. In neurophysiological studies, single-stimulus evoked PRM reflexes are utilized to probe the spinal sensorimotor circuits using specific conditioning-test paradigms [3,8,9,36,37]. The concomitant activation of anterior root efferents would lead to direct M wave-like responses superimposed on the EMG signals of the PRM reflexes, owing to their similar onset latencies [1,2,35]. In interventional studies using tonic transcutaneous spinal cord stimulation [10,11,13,38,39,40,41,42,43], electrical activation of anterior roots would bypass the target spinal circuits and generate continuous contractions of the respectively innervated lower-limb muscles.

We here identified different types of responses evoked by transcutaneous spinal cord stimulation based on their refractory behavior tested by double stimuli. PRM reflexes were characterized by their clear suppression when evoked 35 ms following a preceding activation (e.g., soleus in Figure 2B; hamstrings, tibialis anterior, and soleus in Figure 4) [1,4,6,44]. Other responses had major EMG components that could be evoked in close succession with little to no attenuation (e.g., rectus femoris in Figure 4; cf. Figure 4 in [45]). These responses likely reflected a mixed activation of motor axons in the anterior roots and afferent axons in the posterior roots (Figure A1 and Figure A2) [19,35,46].

The present results suggest that spine alignment affects the preferential site of neural stimulation: in the extended and neutral spine alignment conditions, elicitation of PRM reflexes suggests activation of sensory afferent fibers in the posterior rootlets only; in the flexed spine alignment condition, the decreased suppression of the second response to the double stimuli suggests (co-)activation of motor and sensory fibers of the anterior and posterior roots in the intervertebral foramina. An explanation for this phenomenon could be that the flexed spine alignment condition profoundly alters current flow through the spine and consequently the current acting upon the different neural structures.

Computer simulations showed that the relatively low conductivity of the vertebrae [20] impede the conduction of electrical current across the spine and major current flow directions develop across the ligaments, the dural sac, the intervertebral discs, as well as the intervertebral foramina (Figure A3 and Figure A4) [15,17].

The relative rotations and translations of the rigid vertebrae involved in the flexion and extension of the spine are associated with the deformation of their connecting intervertebral discs and longitudinal and intervertebral ligaments [21]. This results in changes of the volumes occupied by the soft tissues and fluid in the vertebral canal as well as the size of the intervertebral foramina: with sagittal flexion of the spine, the length of the vertebral canal increases because the axes of rotation shift anteriorly towards the intervertebral discs (Figure A5). In addition, all ligaments bordering the canal are stretched [21], the cross-sectional area of the spinal canal at the level of the intervertebral discs increases [47,48], and the vertical and horizontal dimensions of the intervertebral foramina are maximized (Figure A5B). Conversely, with spinal extension, there is a reduction of the cross-sectional area of the spinal canal caused by bulging of the intervertebral discs and an increase in thickness of the ligamenta flava, which are pushed anteriorly by the superior articular processes of the underlying vertebrae [47,48,49]. Additionally, the diameters of the intervertebral foramina decrease with spinal extension as the pedicles come closer together [48,50].

The increased volume of the vertebral canal and increased diameters of the intervertebral foramina in the flexed spine alignment condition could explain the associated reduction in EMG response sizes and increased second-to-first response ratios: stimulation of afferent fibers in the posterior rootlets might have been impeded by decreased current densities in the spinal canal and dural sac, and stimulation of motor fibers might have been augmented by increased channeling of current flow through the intervertebral foramina.

If this theory is true, then the sitting position is the least preferential body position for posterior root stimulation because the pelvis is rotated backwards, resulting in a flattening of the lumbar spine and a strong flexion bias compared to the other studied body positions [51,52]. Even the extended spine condition in sitting shows more relative flexion than the neutral spine condition in the standing and supine body positions [51]. Indeed, the example recording shown in Figure A2 indicates ubiquitous activation of motor fibers in the sitting position. Note that by modeling the body position as random effects, we controlled for such position-specific differences.

The capacity of transcutaneous spinal cord stimulation to activate posterior roots could have been additionally influenced by relative changes in paraspinal electrode location with respect to the spinal cord and roots due to movement of the skin or the spinal cord accompanying sagittal flexion of the spine. However, skin movement was shown to agree with the movement of the underlying spinous processes within about 10% difference [53], and the flexibility of the spinal cord allows it to largely follow the changes of length of the vertebral canal [21]. The influence of potential changes in trajectories and orientations of anterior and posterior roots within the generated electric field remains to be elucidated. The proposed underlying mechanisms remain hypothetical, and further investigations are needed. Specifically, computational studies could help delineate the impact of the various biophysical and anatomical consequences of sagittal spinal flexion on current flow and neural excitation. Detailed computer simulations including digital twin generation may lead to the development of adaptive stimulation methods for reliable afferent fiber stimulation independent from spine alignment conditions.

## 5. Conclusions

Transcutaneous spinal cord stimulation for neurophysiological studies as well as neuromodulative applications relies on the activation of sensory afferent fibers. Here, we showed that afferent fibers can be reliably stimulated with neutral and extended spine alignment conditions in various body positions. In contrast, sagittal flexion of the spine detrimentally impacted the activation of afferent fibers and could result in co-activation of efferent fibers. Thus, transcutaneous spinal cord stimulation should be applied in a body position that allows for stable extended or neutral sagittal spine alignment. Further, the capacity of transcutaneous spinal cord stimulation to recruit afferent fibers should be confirmed in the same body position and spine alignment condition prior to its intended scientific or interventional application.

## Figures and Tables

**Figure 1 jcm-10-05543-f001:**
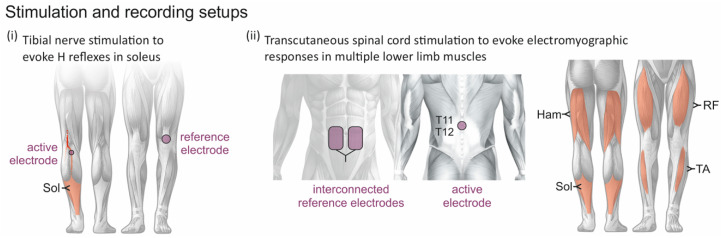
Stimulation and recording setups. (**i**) H reflexes in soleus (Sol) were evoked with an active electrode placed over the posterior tibial nerve in the popliteal fossa and a reference electrode placed proximal to the patella. (**ii**) Posterior root-muscle (PRM) reflexes were elicited bilaterally in the rectus femoris (RF), the hamstrings (Ham) muscle group, tibialis anterior (TA), and Sol by transcutaneous spinal cord stimulation with an active electrode placed over the spine between the T11 and T12 spinous processes and a pair of interconnected reference electrodes over the lower abdomen.

**Figure 2 jcm-10-05543-f002:**
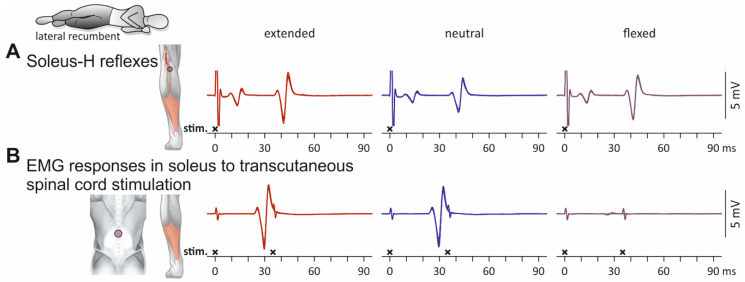
Influence of the spine alignment condition on M waves and H reflexes as well as on electromyographic (EMG) responses to transcutaneous spinal cord stimulation in left soleus: exemplary results. (**A**) EMG responses evoked by tibial nerve stimulation; six stimulus-triggered repetitions displayed superimposed. Traces show stimulation artifacts followed by M waves and (at > 30 ms) H reflexes. (**B**) EMG responses evoked by transcutaneous spinal cord stimulation. Superimposed representation of responses to three single stimuli as well as to three double stimuli with an interstimulus interval of 35 ms. EMG responses to transcutaneous spinal cord stimulation diminished in size during static flexion of the thoracolumbar spine, while M waves and H reflexes did not demonstrate such changes. Black crosses mark times of stimulus application. All recordings derived from the left soleus of one subject in lateral recumbent position, with the spine in extended, neutral, and flexed conditions. Note that amplitudes of tibial nerve and spinal cord stimulation were unchanged across the different spine alignment conditions.

**Figure 3 jcm-10-05543-f003:**
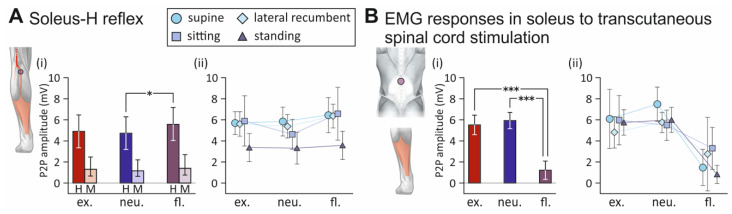
Influence of the spine alignment condition on M waves and H reflexes as well as electromyographic (EMG) responses to transcutaneous spinal cord stimulation in soleus: group results. (**A**) (**i**) Marginal mean EMG peak-to-peak (P2P) amplitudes of H reflexes (H) and M waves (M) elicited across body positions with extended (ex.), neutral (neu.), and flexed (fl.) spine. P2P amplitudes of H reflexes were significantly larger with a flexed than with a neutral spine. (**ii**) P2P amplitudes of H reflexes elicited with different body positions and spine alignment conditions. (**B**) (**i**) Marginal mean P2P amplitudes of soleus responses to transcutaneous spinal cord stimulation across body positions elicited in different spine alignment conditions. P2P amplitudes were significantly smaller with flexed spine alignment compared to extended and neutral alignments. (**ii**) P2P amplitudes of EMG responses elicited with different body positions and spine alignment conditions. Whiskers extend from the lower to the upper limits of the respective 95% confidence intervals. Asterisks denote significant results of pairwise contrasts (*, *p* < 0.05; ***, *p* < 0.0001).

**Figure 4 jcm-10-05543-f004:**
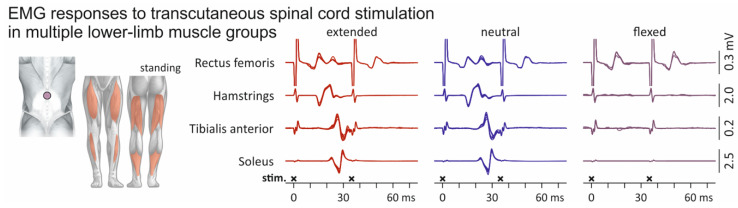
Influence of the spine alignment condition on lower-limb muscle responses evoked by transcutaneous spinal cord stimulation: exemplary results. Electromyographic (EMG) recordings from rectus femoris, hamstrings, tibialis anterior, and soleus. Superimposed representations of three responses to single stimuli (cropped at 35 ms) as well as three responses to double stimuli with an interstimulus interval of 35 ms. Black crosses mark times of stimulus application. All recordings derived from one subject while standing, with the spine in extended, neutral, and flexed alignments. Note that stimulation amplitudes were kept constant across spine alignment conditions.

**Figure 5 jcm-10-05543-f005:**
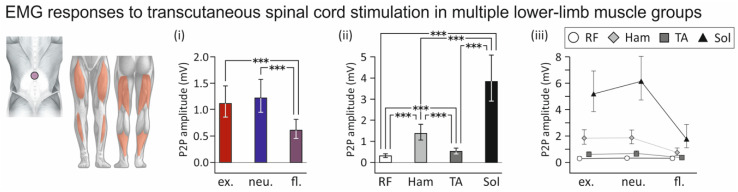
Influence of the spine alignment condition on lower-limb muscle responses evoked by transcutaneous spinal cord stimulation: group results. (**i**) Marginal mean peak-to-peak (P2P) amplitudes of electromyographic (EMG) responses evoked with extended (ex.), neutral (neu.), and flexed (fl.) spine alignment conditions, across body positions (supine, lateral recumbent, sitting, standing) and muscle groups (rectus femoris, RF; hamstrings; Ham; tibialis anterior, TA; soleus, Sol). Response sizes were significantly smaller with spinal flexion than with extension and with neutral spine alignment. (**ii**) Marginal mean P2P amplitudes of responses evoked in RF, Ham, TA, and Sol, across body positions and spine alignment conditions. (**iii**) P2P amplitudes of responses evoked in muscles and with spine alignment conditions. Whiskers extend from the lower to the upper limits of the respective 95% confidence intervals. Asterisks denote significant results of pairwise contrasts (***, *p* < 0.0001).

**Figure 6 jcm-10-05543-f006:**
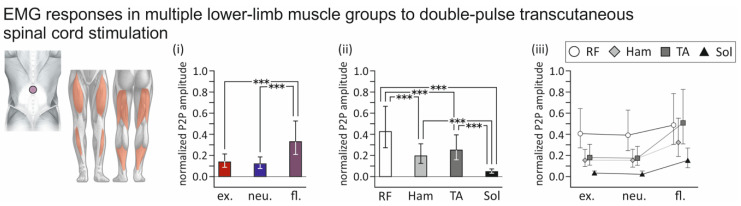
Influence of spine alignment condition on lower-limb muscle responses to double-pulse transcutaneous spinal cord stimulation: group results. (**i**) Marginal mean normalized peak-to-peak (P2P) amplitudes of second-to-first electromyographic (EMG) responses elicited with extended (ex.), neutral (neu.), and flexed (fl.) spine alignment, across body positions and muscle groups. Normalized response sizes were significantly larger with flexed than with extended or neutral spine alignments. (**ii**) Marginal mean normalized P2P amplitudes of responses elicited in rectus femoris (RF), hamstrings (Ham), tibialis anterior (TA), and soleus (Sol), across body positions and spine alignment conditions. (**iii**) Normalized P2P amplitudes of responses elicited in muscle groups and with spine alignments as indicated. Whiskers extend from the lower to the upper limits of the respective 95% confidence intervals. Asterisks denote significant results of pairwise contrasts (***, *p* < 0.0001).

## Data Availability

The data of this study are available in the manuscript.

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
