# Peer review of "Influence of Spine Curvature on the Efficacy of Transcutaneous Lumbar Spinal Cord Stimulation"

_jcm, 2021, doi:10.3390/jcm10235543_

Round 1

Reviewer 1 Report

The objective of this study was to systematically investigate the effects of different postures and spine curvature on electrophysiological characteristics of spinally evoked motor potentials in the lower limb muscles in neurologically intact participants. Authors demonstrate that afferent fibers within the spinal dorsal roots are preferentially activated during both neutral and extended spine alignment conditions in various body positions; whereas, sagittal flexion of the spine impacts activation of afferent fibers and can result in co-activation of efferent fibers in the ventral roots. The hypothesis of the in-vivo study is clear, the experiments are scrupulously conducted, and the data presented are straightforward. The findings are discussed from electrophysiological, orthopedic, and biophysics perspectives, which adds the value on this multidisciplinary study. Practical implications of these findings are irrefragable, as transcutaneous spinal stimulation is gaining attention in various clinical populations as a compelling approach for neurorehabilitation. Indeed, many of the subjects participating in clinical studies using transcutaneous spinal stimulation, experience to a various extent conditions preventing them from fully extended posture. Therefore, this work provide knowledge on the advantages and limitations researchers and clinician should foresee depending on the subjects’ curvature of the spine during spinal stimulation.

My specific minor comments are below:

Title:

I would suggest removing the word “biophysical” from the title, because the study discusses both anatomical and physiological influences of the spine curvature on characteristics of spinally evoked motor potentials.

Methods:

I wonder if it is possible to provide some sort of quantitative description of the spine curvature during the experiments? Although the authors stated that the conditions included “maximum extension” and “maximum flexion”, it is not clear whether the angle of the thoracolumbar spine was kept the same in the tested body positions. For instance, during the tests in standing position, the subjects stood next to a wall for reference, but how then maximum extension was different from neutral position?

The Methods state that “the posterior tibial nerve of the left leg” was stimulated to induce the soleus H-reflex, however, Figures 1i, 2A, and 3A present the right leg.

Results:

Section 3.2: Influence of spine alignment condition on responses in thigh and leg muscles evoked by transcutaneous spinal cord stimulation (also Fig. 5ii): The rational of comparison of muscle effect is not clear. Was this analysis necessary? Is not it expected that the responses will differ across the lower limb muscles given their different composition and functions?

Discission:

Although the findings are carefully discussed from electrophysiology, orthopedics, and biophysics points of view, the proposed mechanisms remain theoretical, and may require further investigation using computational or experimental animal models. Authors could mention this as a limitation of the present study.

Author Response

Thank you for your thoughtful comments and suggestions that helped us improve the manuscript. We addressed all of them in our point-by-point response below and made the corresponding changes in the manuscript.

The objective of this study was to systematically investigate the effects of different postures and spine curvature on electrophysiological characteristics of spinally evoked motor potentials in the lower limb muscles in neurologically intact participants. Authors demonstrate that afferent fibers within the spinal dorsal roots are preferentially activated during both neutral and extended spine alignment conditions in various body positions; whereas, sagittal flexion of the spine impacts activation of afferent fibers and can result in co-activation of efferent fibers in the ventral roots. The hypothesis of the in-vivo study is clear, the experiments are scrupulously conducted, and the data presented are straightforward. The findings are discussed from electrophysiological, orthopedic, and biophysics perspectives, which adds the value on this multidisciplinary study. Practical implications of these findings are irrefragable, as transcutaneous spinal stimulation is gaining attention in various clinical populations as a compelling approach for neurorehabilitation. Indeed, many of the subjects participating in clinical studies using transcutaneous spinal stimulation, experience to a various extent conditions preventing them from fully extended posture. Therefore, this work provide knowledge on the advantages and limitations researchers and clinician should foresee depending on the subjects’ curvature of the spine during spinal stimulation. 

Response: Thank you for the positive assessment of our work.

My specific minor comments are below:

Title:

I would suggest removing the word “biophysical” from the title, because the study discusses both anatomical and physiological influences of the spine curvature on characteristics of spinally evoked motor potentials.

Response: We agree and have removed the word “biophysical” from the title.

Methods:

I wonder if it is possible to provide some sort of quantitative description of the spine curvature during the experiments? Although the authors stated that the conditions included “maximum extension” and “maximum flexion”, it is not clear whether the angle of the thoracolumbar spine was kept the same in the tested body positions. For instance, during the tests in standing position, the subjects stood next to a wall for reference, but how then maximum extension was different from neutral position?

Response: The angle of the thoracolumbar spine in extension, neutral, and flexion, respectively, was not kept the same in the tested body positions. Indeed, body position itself impacts the neutral spine alignment. We comment on this fact in the discussion: “… the sitting position … the pelvis is rotated backwards, resulting in a flattening of the lumbar spine and a strong flexion bias compared to the other studied body positions [44,45]. Even the extended spine condition in sitting shows more relative flexion than the neutral spine condition in the standing and supine body positions [44].”

Our experiments were specifically designed to investigate the effect of a relative change of the spinal alignment and not of absolute values. The different body positions served as controls to exclude body-specific effects, which also includes the degree or extent of relative spinal flexion and extension. This is now stated in 2.2 Investigated body positions and spine alignments: “Maximum extension and flexion were defined as the maximum displacements that the participants felt comfortable to assume. Maximum extension, neutral, and maximum flexion angles, respectively, were not controlled to be the same across body positions. The experiments were designed to investigate the effects of relative changes in spinal alignment.”

The Methods state that “the posterior tibial nerve of the left leg” was stimulated to induce the soleus H-reflex, however, Figures 1i, 2A, and 3A present the right leg.

Response: Thanks for catching this error. Peripheral nerve stimulation was applied on the left leg. We updated the relevant figures.

Results:

Section 3.2: Influence of spine alignment condition on responses in thigh and leg muscles evoked by transcutaneous spinal cord stimulation (also Fig. 5ii): The rational of comparison of muscle effect is not clear. Was this analysis necessary? Is not it expected that the responses will differ across the lower limb muscles given their different composition and functions?

Response: We included the pairwise comparisons of the peak-to-peak amplitudes between muscles for the sake of completeness of the statistical model. We agree that one would expect the amplitudes to be different between muscles, but the statistically significant results and small confidence intervals also demonstrate consistent stimulation conditions across subjects. Furthermore, the fixed effect of muscle must be included in the mixed model to investigate the interaction effect between muscle and spinal alignment and leaving out the description of the fixed effect of muscle would result in an incomplete description of the statistical model. Thus, we preferred to keep the description of the muscle effect.

Discission:

Although the findings are carefully discussed from electrophysiology, orthopedics, and biophysics points of view, the proposed mechanisms remain theoretical, and may require further investigation using computational or experimental animal models. Authors could mention this as a limitation of the present study.

Response: We agree with the comment and added a statement at the end of the discussion noting that the mechanisms discussed are hypothetical and need further investigation (see below). We also made sure to use the conjunctive when speculating about the mechanisms throughout the discussion.

Statement added to the discussion: “The proposed underlying mechanisms remain hypothetical and further investigations are needed. Specifically, computational studies could help to delineate the impact of the various biophysical and anatomical consequences of sagittal spinal flexion on current flow and neural excitation. Detailed computer simulations including digital twin generation may lead to the development of adaptive stimulation methods for reliable afferent fiber stimulation independent from spine alignment conditions.” 

Reviewer 2 Report

In the present manuscript, Binder et al investigate the effect of spine curvature in different body positions during both transcutaneous lumbar spinal cord stimulation and H-reflex testing. This is a well-written manuscript that investigates an interesting topic, performs robust statistical testing, and fills a knowledge gap in the neuromodulation field. However, there are some methodological and interpretation questions that need to be answered prior

Major Comments

  • In addition to the influence of spine curvature and body position on the efficacy of transcutaneous lumbar spinal cord stimulation, the body habitus of the subjects may contribute to biophysical changes in stimulation. Was the BMI of the subjects in the current study similar? If not, did the authors observe any changes in EMG signals between subjects of dissimilar body habitus, and how would that correlate with the results shown with regard to spine curvature?
  • What was the range of spinal stimulation amplitudes used to match the H-reflex peak-to-peak measurements? Did this differ greatly between subjects or between positions? This could be very useful information for future studies investigating stimulation in any of the presently examined body positions.
  • The data shown for the flexed condition seems to show similar trends to the prone body position in one of the authors’ previous publications (Danner et al 2016), which indicated an increased recruitment of anterior root fibers compared to sitting and standing conditions. Based on the present data, do the authors believe that the mechanism of action is similar? Additionally, did the authors attempt the prone condition during any of the current studies? It would be interesting to determine if the flexed, neutral, or extended condition in the prone condition resulted in any differences in PRM responses.
  • In the discussion, could the authors further describe what the results may imply for functional and clinical studies moving forward? In the introduction, the authors describe that transcutaneous lumbosacral stimulation has been used as a neuromodulatory tool for spinal cord injury and multiple sclerosis, and recommend applying transcutaneous stimulation in body positions that allow for a neutral or extended spine in the abstract. However, why the flexed condition is inferior is not elaborated on in great detail. From a high-level view, how could the results shown here that PRM responses while flexed are decreased affect future studies and technological development within the neuromodulation field?

Minor Comments

  • The introduction only discusses afferent pathways for transcutaneous stimulation, which is the primary path of activation. However, the authors later on in the results and discussion sections introduce the fact that transcutaneous lumbosacral stimulation can also directly activate anterior roots, which is critical to understanding the results shown here. Therefore, the introduction should reflect that transcutaneous lumbosacral stimulation primarily activates afferent pathways, but may engage other sensorimotor circuitry to introduce this concept prior to the results section.
  • Line 65, “collapse” is a strange term for a scientific publication. Terms like “decrease” or “diminish” seem more appropriate. This is used in other parts of the manuscript as well and should be revised.
  • Line 85, How was maximum extension and maximum flexion of the thoracolumbar spine defined? Was this based on the maximum displacement that the subject could tolerate or was this defined on some biomechanical standard?
  • Line 89-91, “In the supine position, the examination table was adjusted, and support was provided to stabilize the different spine alignments.” What type of examination table was used to adjust the spine alignments while the subjects were supine? In general, it is somewhat hard to visualize the differences between the flexed/neutral/extended positions in the four different conditions. The diagrams in Figure A2 were very helpful in understanding the sitting positions. Perhaps something similar could be created for all four conditions?
  • How was the value of 35ms for the interstimulus interval chosen? Based on Figure 2, it seems like the second stimulus occurs during the initial evoked response, which may interfere with some of the analysis for that response.
  • Did the subjects self-report any changes in sensation of the stimulation with the decreased EMG response during transcutaneous stimulation in the flexed condition?
  • There is a typo in Figure 6 title: “transcuatenous”

Author Response

Thank you for your thoughtful comments and suggestions that helped us improve the manuscript. We addressed all of them in our point-by-point response below and made the corresponding changes in the manuscript.

In the present manuscript, Binder et al investigate the effect of spine curvature in different body positions during both transcutaneous lumbar spinal cord stimulation and H-reflex testing. This is a well-written manuscript that investigates an interesting topic, performs robust statistical testing, and fills a knowledge gap in the neuromodulation field. However, there are some methodological and interpretation questions that need to be answered prior

Major Comments

In addition to the influence of spine curvature and body position on the efficacy of transcutaneous lumbar spinal cord stimulation, the body habitus of the subjects may contribute to biophysical changes in stimulation. Was the BMI of the subjects in the current study similar? If not, did the authors observe any changes in EMG signals between subjects of dissimilar body habitus, and how would that correlate with the results shown with regard to spine curvature?

Response: This is an interesting question that we are unfortunately not able to answer with the current set of data. All but one subject were of normal weight (mean: 21.8; range: 18.8–25.7) and thus no statistical comparison is possible. We added information on the BMI of the participants to the methods section. 

What was the range of spinal stimulation amplitudes used to match the H-reflex peak-to-peak measurements? Did this differ greatly between subjects or between positions? This could be very useful information for future studies investigating stimulation in any of the presently examined body positions.

Response: This would be an interesting question to investigate. Yet, we did not design the experiments to investigate differences between body positions (the different body positions were used to ensure that the effect of spinal curvature was not specific to a given body position) and comparing the stimulation intensities between body positions would not lead to reliable results. As described in the methods, matching the amplitude of the EMG responses to transcutaneous spinal stimulation in the left soleus to those of the H-reflex in the same muscle was not the only criterion used to determine the stimulation intensity for each body position. We also required EMG responses to be elicited in all recorded muscles bilaterally and only chose stimulation intensities that the participants found comfortable. Furthermore, the stimulation intensity was limited by the stimulator to 125 mA, which resulted in a ceiling-effect. After removal of the confounded data, the mean and standard deviation of the intensities were 50.7±14.9 mA in supine position (n=7), 79.4±22.2 mA during standing (n=5), 74.7±30.7 mA while sitting (n=6), and 67.8±22.3 mA in the lateral recumbent position (n=6). There were no significant differences (one-way ANOVA, P=0.196). A properly designed study would be needed to assess the differences in stimulation intensity to elicit comparable EMG responses in different body positions; no conclusions can be drawn from the current data set.

The data shown for the flexed condition seems to show similar trends to the prone body position in one of the authors’ previous publications (Danner et al 2016), which indicated an increased recruitment of anterior root fibers compared to sitting and standing conditions. Based on the present data, do the authors believe that the mechanism of action is similar?

Response: According to our unpublished preliminarily MRI data, a change from supine to prone body position likely does not result in a relative sagittal flexion of the spine, but rather in a more extended sagittal spine alignment. This suggests that it is unlikely that the same mechanism of action underlies the coactivation of motor fibers in prone body position and with flexed spine alignment. Yet, because we couldn’t find any reliable data in the literature on spinal alignment in prone position in individuals without spinal deformities, we refrain from adding this speculation to the main text.

Additionally, did the authors attempt the prone condition during any of the current studies? It would be interesting to determine if the flexed, neutral, or extended condition in the prone condition resulted in any differences in PRM responses.

Response: We here focused on positions in which recent neuromodulation or neurophysiological studies had applied transcutaneous spinal cord stimulation. We did not test the effect of spinal alignment on reflex responses elicited in the prone position, since we had assumed that this position would be unfavorable for transcutaneous spinal cord stimulation applications.

In the discussion, could the authors further describe what the results may imply for functional and clinical studies moving forward? In the introduction, the authors describe that transcutaneous lumbosacral stimulation has been used as a neuromodulatory tool for spinal cord injury and multiple sclerosis, and recommend applying transcutaneous stimulation in body positions that allow for a neutral or extended spine in the abstract. However, why the flexed condition is inferior is not elaborated on in great detail. From a high-level view, how could the results shown here that PRM responses while flexed are decreased affect future studies and technological development within the neuromodulation field?

Response: We are very grateful for this important feedback. We are now addressing this point already in the revised introduction:
“To reliably apply transcutaneous lumbar spinal cord stimulation, posterior root afferent fibers must be activated selectively and consistently. Indeed, although the stimulation seems unspecific, the biophysical properties of the lumbosacral spinal cord and its surrounding spinal roots turn the stimulation to a rather selective recruitment of large-diameter posterior root fibers [15–18]. Another possible target of the stimulation are anterior roots containing the axons of the spinal motoneurons [15,16]. In neurophysiological studies, their (co-)activation would evoke direct M-wave like responses [1,2] that bypass the spinal sensorimotor circuits, and in neuromodulation approaches, continuous contractions of the respectively innervated muscles would be generated, impeding the intended effects. …”

We have further added the following paragraph to the discussion:
“Mixed activation of sensory posterior root and motor anterior root fibers would be disadvantageous for both major applications of transcutaneous spinal cord stimulation, i.e., neurophysiological as well as interventional studies. In neurophysiological studies, single-stimulus evoked PRM reflexes are utilized to probe the spinal sensorimotor circuits using specific conditioning-test paradigms [3,8,9,37,38]. The concomitant activation of anterior root efferents would lead to direct M-wave like responses superimposed on the EMG signals of the PRM reflexes, owing to their similar onset latencies [1,2,36]. In interventional studies using tonic transcutaneous spinal cord stimulation [10,11,13,39–44], electrical activation of anterior roots would bypass the target spinal circuits as well as generate continuous contractions of the respectively innervated lower-limb muscles.”

Minor Comments

The introduction only discusses afferent pathways for transcutaneous stimulation, which is the primary path of activation. However, the authors later on in the results and discussion sections introduce the fact that transcutaneous lumbosacral stimulation can also directly activate anterior roots, which is critical to understanding the results shown here. Therefore, the introduction should reflect that transcutaneous lumbosacral stimulation primarily activates afferent pathways, but may engage other sensorimotor circuitry to introduce this concept prior to the results section. 

Response: We agree and added a short paragraph introducing the possibility of anterior root activation (see above).  

Line 65, “collapse” is a strange term for a scientific publication. Terms like “decrease” or “diminish” seem more appropriate. This is used in other parts of the manuscript as well and should be revised.

Response: Thank you for this comment. We made the appropriate changes.

Line 85, How was maximum extension and maximum flexion of the thoracolumbar spine defined? Was this based on the maximum displacement that the subject could tolerate or was this defined on some biomechanical standard?

Response: Maximum extension and flexion were defined as the maximum displacement that was comfortable for the participants. We added an explanation to the methods section: 
“Maximum extension and flexion were defined as the maximum displacements that the participants felt comfortable to assume. Maximum extension, neutral, and maximum flexion angles, respectively, were not controlled to be the same across body positions. The experiments were designed to investigate the effects of relative changes in spinal alignment.”

Line 89-91, “In the supine position, the examination table was adjusted, and support was provided to stabilize the different spine alignments.” What type of examination table was used to adjust the spine alignments while the subjects were supine? In general, it is somewhat hard to visualize the differences between the flexed/neutral/extended positions in the four different conditions. The diagrams in Figure A2 were very helpful in understanding the sitting positions. Perhaps something similar could be created for all four conditions?

Response: The table was a standard examination table, and spine alignments in the supine position were supported by the use of therapeutic pillows. Consistent spine alignments across participants were ensured based on the expertise of an orthopedic surgeon (co-author A.R.), rather than by the use of specific support technology.

We have slightly adjusted the methods section:
            “…therapeutic pillows were used to support and stabilize the different spine alignments.”

How was the value of 35ms for the interstimulus interval chosen? Based on Figure 2, it seems like the second stimulus occurs during the initial evoked response, which may interfere with some of the analysis for that response.

Response: The interstimulus interval was chosen because it generally leads to a complete depression of the PRM reflexes. We have previously used even 30 ms (Danner et al. 2016). We visually inspected all recordings to ensure that the artifact of the second pulse did not contribute to the calculation of the peak-to-peak amplitude. Furthermore, we applied 3 double pulses and 3 single pulses. While there was no interference of the second stimulation, if it had occurred, we could have only analyzed the responses to the single stimuli. We now mention the visual inspection of the data in the methods.

Did the subjects self-report any changes in sensation of the stimulation with the decreased EMG response during transcutaneous stimulation in the flexed condition?

Response: None of the subject self-reported any changes in sensation between the spinal alignment conditions.

There is a typo in Figure 6 title: “transcuatenous”

Response: Thanks for catching the typo.

Round 2

Reviewer 2 Report

The authors provided sufficient responses to my previous comments and addressed the relevant issues within the manuscript appropriately. I congratulate the authors on an excellent manuscript and recommend acceptance.